# Low recruitment drives the decline of red porgy (*Pagrus pagrus*) along the southeast USA Atlantic coast: Inferences from fishery-independent trap and video monitoring

**Nathan M. Bacheler**[1]*, **Nikolai Klibansky**[1], **Walter J. Bubley**[2], **Tracey I. Smart**[2]

**1** Southeast Fisheries Science Center, National Marine Fisheries Service, Beaufort, North Carolina, United States of America, **2** South Carolina Department of Natural Resources, Marine Resources Research Institute, Charleston, South Carolina, United States of America

* nate.bacheler@noaa.gov

## Abstract

Red porgy (*Pagrus pagrus*) is a reef-associated, economically-important, winter-spawning, protogynous Sparidae species that appears to have declined in abundance in recent years along the southeast United States Atlantic coast. We used spatially-explicit generalized additive models built with fishery-independent chevron trap (1990–2021) and video data (2011–2021) to quantify the ways in which red porgy relative abundance and mean size varied across temporal, spatial, environmental, and habitat variables. Mean red porgy relative abundance from traps declined by 77% between 1992 and 2021, and declines were similarly large (69%) on video between 2011 and 2021. The largest two-year decline in relative abundance occurred early in the COVID-19 pandemic (2019–2021)– 32% in traps and 45% on video–despite already low abundance. Highest red porgy relative abundance from traps and video occurred in deep areas (i.e., 60–100 m) between southern North Carolina and north Georgia, and red porgy preferred low relief but continuous hardbottom habitats (i.e., pavement). We confirmed recent low recruitment of red porgy in the region based on the large increase in mean length (29%) and severe (~99%) declines of juvenile red porgy caught over the 32-year trap survey. Evidence suggests that recruitment failure is partially or mostly responsible for red porgy abundance declines, and, moreover, the regulation of harvest is unlikely to achieve sustainable management goals until recruitment increases.

## Introduction

Accurate and precise information about relative abundance is a critical component of fish stock assessments [1]. Typically, there are two types of relative abundance estimates that can be included in stock assessments: those from fishery-dependent or fishery-independent data sources. Fishery-dependent data includes catch-per-unit-effort information from the fishing industry itself and are generally less expensive to collect, but there are many examples where fishery-dependent catch-per-unit-effort data does not reflect relative abundance due to

**Data Availability Statement:** There are serious ethical concerns with making these potential fishing points (precise latitudes and longitudes)

freely available to the public. Making these locations public is a significant conservation concern. Data can be requested from either the Corresponding Author, or Walter J. Bubley, a coauthor on this work, or J. Kevin Craig (kevin.craig@noaa.gov) who oversees teh data collection program.

**Funding:** The survey is funded by the U.S. National Marine Fisheries Service, but the authors received no specific funding for the analyses. The funders had no role in study design, data collection and analysis, decision to publish, or preparation of the manuscript.

**Competing interests:** The authors have declared that no competing interests exist.

hyperstability, spatially-variable fishing effort, time or area fishery closures, or unreliable data collection [2–4].

Fishery-independent data are collected from scientific surveys and are typically not subject to the same drawbacks as fishery-dependent data, so the trends from survey data are considered more reliable than from fishery-dependent sources [5,6]. Trawls are the most common survey gear for fishes occupying softbottom habitats, but trawls cannot be used to sample fish associated with rugose reef habitats. Instead, reef-associated fish species have been surveyed with a wide variety of alternative gears such as underwater visual census, hook-and-line, longlines, crewed or uncrewed underwater vehicles, traps, or underwater video [7–12]. Surveys for reef-associated fishes using these diverse gears have provided valuable information on trends in relative abundance that have been used to support efforts to sustainably manage myriad important species.

Red porgy (*Pagrus pagrus*) is a reef-associated fish species that has been surveyed scientifically for decades using traps and more recently video along the southeast United States Atlantic coast (hereafter, SEUS) [13]. Red porgy is a protogynous hermaphroditic Sparidae species that has been targeted by fishers for many years in the SEUS [14,15]. Vaughan and Prager [16] documented a substantial increase in fishing mortality and a decline in spawning potential ratio for red porgy in the SEUS between the 1970s and 1990s, with resulting declines in biomass of 89% and recruitment of 97% over the same time period. The most recent stock assessment of red porgy, which used long-term fishery-independent trap and video data, indicated that the stock was still overfished and undergoing overfishing [17]. Moreover, Smart et al. [18] analyzed long-term trap data to show that adult and recruit abundances were both declining throughout the 2010s. Because of the multispecies nature of the fishery for reef fishes [18] and the high spatial overlap of red porgy with many other targeted reef fishes in the SEUS [19], bycatch and discard mortality continue to remove red porgy from the population even with strict management regulations. Expected recovery has not been seen in the red porgy population, indicating factors besides fishing pressure could be at play.

Here we provide detailed analyses of long-term (32-year) fishery-independent trap and video survey data for red porgy in the SEUS to address the following three specific objectives. Our first objective was to quantify the temporal changes in relative abundance of red porgy in the SEUS using long-term, spatially-explicit trap (1990–2021) and video data (2011–2021). Our second objective was to determine the spatial patterns in relative abundance of red porgy in the SEUS and how red porgy relative abundance varied across environmental and habitat variables. Given the observed declines of red porgy over time (see *Results*), our third objective was to determine whether declines in abundance were partially or completely due to persistently low recent recruitment. These results build upon the previous work of Smart et al. [18] and others to improve our understanding of the temporal and spatial dynamics of red porgy in the SEUS as managers evaluate novel ways to manage this species, given that traditional management measures like size and bag limits do not appear to be achieving the desired results.

## Materials and methods

### Ethics statement

Data collected for this study was authorized via Scientific Research Permits issued by the Administrator of the Southeast Regional Office of the National Marine Fisheries Service, National Oceanic and Atmospheric Administration, United States Government. These Scientific Research Permits covered all areas and organisms sampled in this study. All research followed the guidelines of the U.S. Government Principles for the Utilization and Care of

Vertebrate Animals Used in Testing, Research, and Training (https://olaw.nih.gov/policies-laws/phs-policy.htm).

## Study area

The spatial footprint for this study extended from Cape Hatteras, North Carolina, in the north to St. Lucie Inlet, Florida, in the south and included continental shelf, shelf-break, and upper slope habitats (total area ~100,000 km$^2$; Fig 1). The seafloor in this region is primarily composed of unconsolidated mud and sand substrate, but hardbottom temperate reef habitats are interspersed throughout the region [20]. These patches of reef habitat are highly variable, ranging from flat limestone pavement often covered by a veneer of sand to high-relief rocky ledges [21]. Red porgy associate with these patches of hardbottom [18,22] and were the focus of sampling in this study.

## Data collection

We used chevron trap and video data collected by the Southeast Reef Fish Survey (SERFS) for this study. SERFS is composed of three different entities that work together collaboratively to sample reef fishes in the SEUS. The first is the Marine Resources Monitoring, Assessment, and Prediction (MARMAP) program, housed at the South Carolina Department of Natural Resources, which has been sampling reef fishes with chevron traps in the region since 1990. The second is the Southeast Area Monitoring and Assessment Program, South Atlantic Region (SEAMAP-SA) Reef Fish Complement, which is also based at SCDNR and has sampled in the SEUS since 2009. The third is the Southeast Fishery-Independent Survey of the National Marine Fisheries Service, which was created in 2010 to increase sampling in the region and incorporate underwater video into the overall SERFS survey.

SERFS used a simple random sampling design to select stations for sampling each year from a sampling frame composed of known hardbottom habitat. A portion of stations from the sampling frame were selected for sampling each year, and the number of stations available in the sampling frame has increased over time as more patches of hardbottom habitats have been discovered. Most of the stations included in our analyses were randomly selected, but some stations not selected for sampling were sampled opportunistically in order to increase the sampling efficiency during research cruises. Moreover, some new hardbottom stations were discovered and sampled each year and were included in our analyses if hardbottom was present. Sampling occurred on the R/V *Palmetto*, R/V *Savannah*, NOAA Ship *Nancy Foster*, NOAA Ship *Pisces*, and NOAA Ship SRVx *Sand Tiger* during daylight hours between the spring and fall each year (Table 1).

Baited chevron traps (aka arrowhead or Madeira traps) have been deployed by SERFS since 1990 in a standardized way to sample reef fish species in the SEUS. Chevron traps used in this study were shaped like an arrowhead, were constructed from 3.4 × 3.4 cm wire mesh, and were 1.7 × 1.5 × 0.6 m in size, with a total volume of 0.91 m$^3$ [7,23]. Chevron traps were baited with 24 menhaden (*Brevoortia* spp.), four on each of four stringers and eight placed loosely inside the trap. Each trap was deployed individually and was attached with a line to two surface buoys for retrieval. Target soak time was 90 min for each trap, and the minimum distance between traps within a given year was 200 m to provide independence between traps. Red porgy trap catch-per-unit-effort was calculated as the number of individuals caught per trap. Chevron trap samples were removed from our analyses if the validity of the sample was questionable for any reason such as evidence that the trap was dragging or bouncing or any information was missing for the sample.

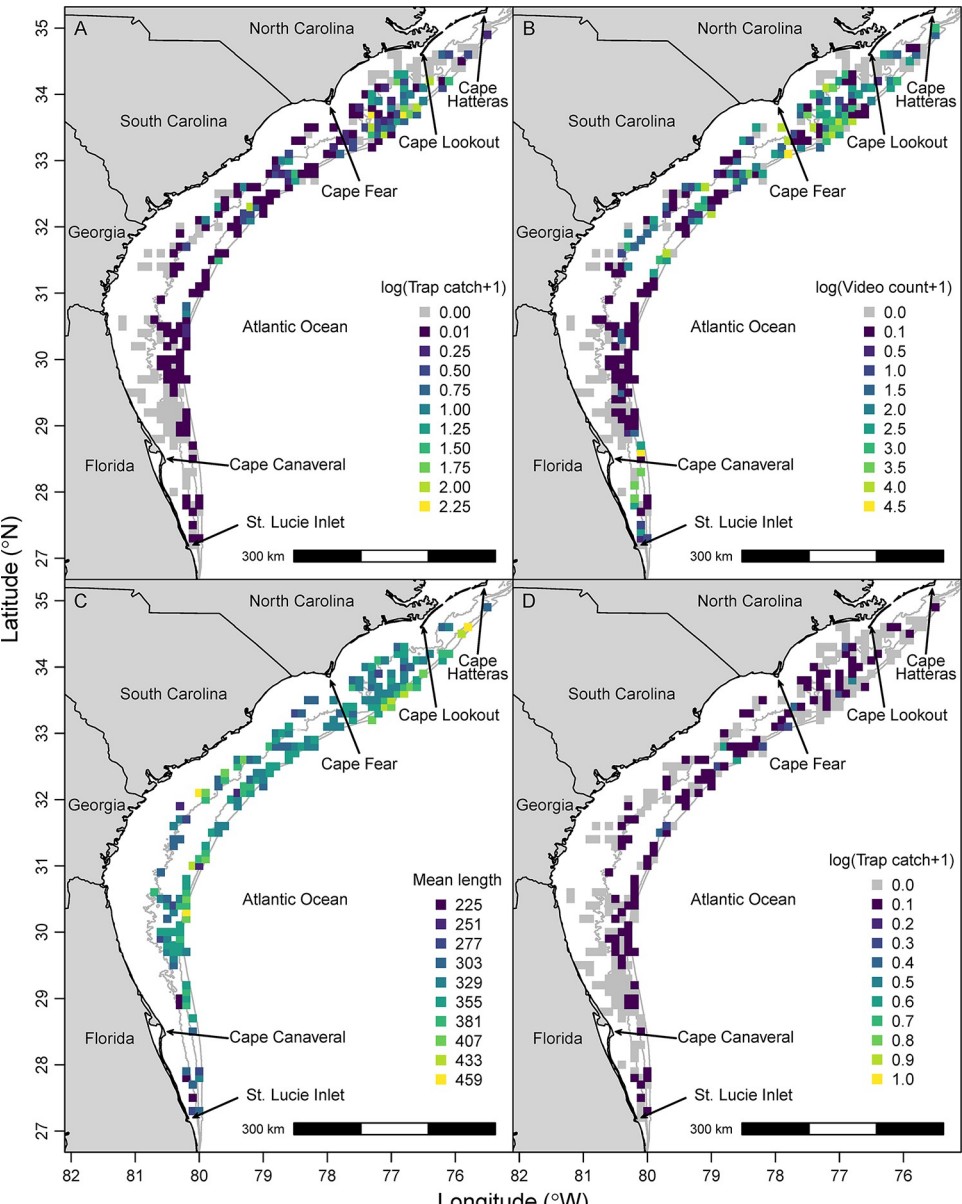

**Fig 1. Heat maps of red porgy (*Pagrus pagrus*) trap catch, video counts, total length, and juvenile trap catches.** (A) Mean log-transformed trap catch of all red porgy (*Pagrus pagrus*; 1990–2021), (B) mean log-transformed video SumCount of all red porgy (2011–2021), (C) mean total length (mm) of red porgy caught in traps (1990–2021), and (D) mean log-transformed trap catch of juvenile (< 290 mm total length) red porgy (1990–2021). Gray cells show areas where sampling occurred but no red porgy were caught in traps or observed on video, and colored cells show mean values across all traps or videos within that cell.

SERFS began attaching underwater video cameras to chevron traps region-wide in 2011 to collect additional information on reef fish temporal and spatial distributions [24]. Two cameras have been attached to chevron traps since 2011: one over the trap mouth that was used to count fish and quantify habitat and one over the trap nose to quantify habitat in the opposite direction. In 2011–2014, Canon Vixia HF-S200 video cameras in Gates HF-S21 housings were attached over the trap mouth to count fish and either GoPro Hero or Nikon Coolpix S210/ S220 were attached over the trap nose. From 2015–2021, GoPro Hero 3+/4 cameras were used

**Table 1. Annual sampling information for the 32-year chevron trap and video survey by the Southeast Reef Fish Survey along the southeast United States Atlantic coast.**

| Year | Trap *N* | Video *N* | Mean depth (m; range) | Mean latitude (˚N; range) |
|---|---|---|---|---|
| 1990 | 302 | 0 | 33 (16–91) | 32.5 (30.4–33.8) |
| 1991 | 257 | 0 | 34 (16–93) | 32.6 (30.8–34.6) |
| 1992 | 286 | 0 | 33 (16–59) | 32.8 (30.4–34.3) |
| 1993 | 354 | 0 | 35 (16–94) | 32.5 (30.4–34.3) |
| 1994 | 332 | 0 | 39 (15–94) | 32.3 (30.7–33.8) |
| 1995 | 291 | 0 | 35 (16–59) | 32.2 (29.9–33.7) |
| 1996 | 318 | 0 | 36 (14–94) | 32.5 (30.0–34.3) |
| 1997 | 327 | 0 | 38 (15–94) | 32.2 (27.9–34.6) |
| 1998 | 341 | 0 | 38 (15–93) | 32.2 (27.4–34.6) |
| 1999 | 210 | 0 | 36 (15–57) | 31.9 (27.3–34.6) |
| 2000 | 248 | 0 | 37 (15–91) | 32.2 (29.0–34.3) |
| 2001 | 223 | 0 | 40 (15–91) | 32.3 (27.9–34.3) |
| 2002 | 221 | 0 | 39 (15–94) | 31.8 (27.9–34.0) |
| 2003 | 202 | 0 | 41 (16–92) | 32.0 (27.4–34.3) |
| 2004 | 248 | 0 | 41 (15–93) | 32.3 (30.0–34.0) |
| 2005 | 276 | 0 | 40 (16–71) | 32.0 (27.3–34.3) |
| 2006 | 274 | 0 | 39 (16–94) | 32.3 (27.3–34.4) |
| 2007 | 310 | 0 | 40 (16–94) | 32.1 (27.3–34.3) |
| 2008 | 276 | 0 | 40 (15–92) | 32.1 (27.3–34.6) |
| 2009 | 404 | 0 | 37 (15–93) | 32.2 (27.3–34.6) |
| 2010 | 762 | 0 | 39 (15–93) | 31.3 (27.3–34.6) |
| 2011 | 721 | 580 | 41 (15–94) | 30.9 (27.2–34.5) |
| 2012 | 1173 | 1083 | 41 (15–98) | 31.9 (27.2–35.0) |
| 2013 | 1356 | 1221 | 38 (15–98) | 31.3 (27.2–35.0) |
| 2014 | 1469 | 1382 | 39 (16–98) | 31.9 (27.2–35.0) |
| 2015 | 1456 | 1405 | 40 (15–96) | 31.9 (27.3–35.0) |
| 2016 | 1482 | 1404 | 41 (16–99) | 32.1 (27.2–35.0) |
| 2017 | 1517 | 1424 | 41 (15–99) | 32.0 (27.2–35.0) |
| 2018 | 1727 | 1654 | 41 (16–98) | 32.0 (27.2–35.0) |
| 2019 | 1653 | 1545 | 40 (14–98) | 32.0 (27.2–35.0) |
| 2020 | 0 | 0 | - | - |
| 2021 | 1904 | 1383 | 40 (16–99) | 31.9 (27.2–35.0) |
| Overall | 20920 | 13081 | 14–99 | 27.2–35.0 |

Trap *N* = number of chevron trap samples included in the analyses each year and Video *N* = number of video samples included in the analyses each year.

over the trap mouth and nose. Videos were removed from our analyses if they were out of focus, dark, or did not record for any reason. Thus, video data were available from 2011 to 2021 and trap data were available from 1990 to 2021; note that no sampling occurred in 2020 due to the coronavirus-19 pandemic.

Relative abundance of red porgy on video was calculated using a derivation of the Mean-Count approach [25]. Fish are commonly counted on videos using the *MaxN* approach [26], which is the maximum number of individuals of a given species observed in a single video frame, but *MaxN* can be nonlinearly related to true abundance for some species [25,27]. Schobernd et al. [25] instead recommended using the MeanCount metric, which is the mean number of individuals of a given species counted across a series of snapshots within a video. For

our study, we used a closely related metric, SumCount, which was the sum of individuals of a given species observed across a series of snapshots; SumCount is exactly proportional to Mean-Count when the number of frames examined was the same [28]. SumCount was used instead of MeanCount because some of the error distributions we examined (e.g., negative binomial, Poisson) required count data. We began reading videos 10 minutes after the trap landed on the bottom and counted red porgy on individual frames spaced every 30 seconds over a 20-min interval of time, for a total of 41 frames read in total. Any video with fewer than 41 frames read was excluded from the analyses.

We conducted a camera calibration study in 2014 to account for the two camera types used to count red porgy in our study. A total of 143 traps were deployed with Canon and GoPro cameras attached side-by-side over the trap mouth, and those resulting videos were read using SumCount at the exact same times on the two different camera types. Red porgy were observed on 24 pairs of calibration videos, and, using a linear model, we determined that the Canon cameras observed a mean of 41.8% fewer red porgy than GoPro cameras, similar to the difference in field-of-view between cameras. To account for the camera change, we therefore reduced the video index in 2015–2021 by 41.8% to make years 2015–2021 comparable to and consistent with data in 2011–2014.

We also recorded several variables at each station sampled in our study. We used the ship's global positioning system to determine the latitude and longitude of each sample and used the vessel's echosounder to determine depth (m). Bottom water temperature (˚C) was measured using a "conductivity-temperature-depth" cast deployed at each group of simultaneously deployed traps. Trap soak time was measured as the elapsed time between the trap being deployed off the back deck of the ship to the beginning of the trap retrieval process. Additional variables were estimated from underwater videos. We recorded two habitat variables from each of the two cameras attached to traps in our study [29]. The first was percent of the visible substrate that was consolidated hardbottom, and a mean value was estimated across the two cameras. The second was substrate relief, which was the maximum substrate relief visually estimated from either camera and was measured in three categories: low (< 0.3 m), moderate (0.3–1.0 m), or high (> 1.0 m). We also recorded the current direction as "away," "sideways," or "towards" based on the movement of particles in the water relative to the video camera over the trap mouth. Last, water clarity was classified as "high" if the horizon was visible in the distance, "moderate" if the substrate could be seen but not the horizon, and "low" if the substrate could not be seen. Samples were excluded from analyses if any of these values were missing or unknown.

## Temporal and spatial patterns of red porgy

Our first objective was to estimate relative abundance of red porgy in the SEUS over time using chevron trap (1990–2021) and underwater video (2011–2021) data. To address this objective, we first summarized observed (unstandardized) trap and video data. For each year of the study, we calculated the proportion of traps that caught red porgy and the proportion of videos in which red porgy were observed (hereafter, "proportion positive"). Second, we calculated the mean trap catch (number of individuals per trap) and mean video SumCount of red porgy for each year of the study. Third, to examine spatial patterns, we summarized observed trap catches and video SumCounts of red porgy across the study area. To do this, we created a 0.1 × 0.1˚ grid over the study area, and plotted mean values within each cell. One potential downside of analyzing observed data is that any changes in the annual spatial or temporal footprint of sampling or environmental conditions over time could be confounded with annual changes in red porgy relative abundance [4,29].

To address this concern, we also developed spatially explicit generalized additive models (GAMs) to standardize trap catches or video counts with a number of predictor variables that were hypothesized to influence red porgy trap catch or video counts. GAMs are regression models that use local smoothers to fit potentially nonlinear relationships between response and predictor variables, which are commonly observed in ecology [30]. Another benefit of GAMs is that they can fit a number of different error distributions [31].

Our first GAM related the trap catch of red porgy to six predictor variables. These predictor variables were year (*year*; 1990–2021), depth (*depth*, m), bottom water temperature (*temp*, ˚C), day of the year (*doy*), trap soak time (*dur*; min), and position (*pos*). We excluded trap samples with soak times less than 50 min and greater than 150 min, as well as samples greater than 100 m deep, due to low sample sizes. Position was a bivariate smooth predictor that was created using the latitude and longitude of the sample [32]. No predictor variables exhibited multicollinearity based on the variance inflation factors [33].

Our first trap catch GAM was coded as:

$$y = f(year) + s(depth) + s(temp) + s(doy) + s(dur) + s(pos),    \tag{1}$$

where *y* is the trap catch of red porgy, *year* is year of the sample, *depth* is bottom depth of the sample, *temp* is the bottom temperature, *doy* is day of the year, *dur* is trap soak time, *pos* is the two-dimensional position variable, *f* is a categorical function, and *s* is a cubic spline (smoothed) function. All GAMs were coded and analyzed in R version 4.1.1 [34] using the mgcv library 1.8–23 [35].

We compared numerous error distributions and data transformations to identify models that exhibited the best fit. We examined Gaussian, gamma, Tweedie, Poisson, and negative binomial models with one of three possible data transformations of the response variable: none, fourth root, or log. The best fitting trap catch model, based on various model diagnostics using the "gam.check" function, was a Gaussian error distribution with a log-transformation of red porgy trap catch, so it was selected and used. This and all subsequent GAMs met assumptions of normality and constant variance. Note that the degree of flexibility in smoothed covariates was determined by the built-in algorithm in the mgcv library.

We then compared the full model containing all six predictor variables to reduced models that contained fewer predictor variables. We used Akaike's information criterion (AIC) for model selection, which identified the most parsimonious model that maximizes fit with the fewest parameters possible [36]. Models with the lowest AIC values were considered the best model in the set; we used ΔAIC for all reporting, which was a measure of each model relative to the best model in the set. Best models had a ΔAIC = 0, and other models have ΔAIC > 0. Generally, models with ΔAIC values of less than 2 are similarly supported by the data, and those with ΔAIC values of greater than 2 have less support [36].

We also examined relationships between response and individual predictor variables that were included in the final (best) model. We were principally interested in extracting the year effect from the model, but we addressed our second objective by quantifying how red porgy trap catches were related to other predictor variables to determine habitat use, spatial distribution patterns, and how their catches might be influenced by soak time. To extract predictor variable plots, we used the best GAM (based on ΔAIC) to predict red porgy trap catches at average values of all other predictor variables. Year is a categorical variable, so when predicting the effects of other predictor variables, the year 2000 was chosen (haphazardly), but note that predictor variable effects were unaffected by the choice of year. These predictor variable plots show mean effects and 95% confidence intervals. We also used our spatially explicit GAM to predict red porgy trap catches across the study area at a spatial resolution of 90 m. For this

plot, we used the latitude, longitude, and depth of each cell and predicted trap catches using average values for all other predictor variables in the model. This plot predicts red porgy relative abundance across the SEUS.

We next determined the relative influence of each individual predictor variable to explaining variability in red porgy trap catch. To accomplish this, we estimated the deviance explained by GAMs that included each predictor variable by itself, so that objective comparisons could be made about the relative importance of each predictor variable in the best model.

Our second GAM related video SumCount of red porgy to nine predictor variables. We pursued video analyses here in addition to the trap analyses above because, despite the time series being shorter for video, red porgy are observed on video more frequently than they are caught in chevron traps [24], so video-based relative abundance may be estimated with greater precision than with traps. All predictor variables included in the trap model above were included in the video model except trap soak time (*dur*), since that predictor was not relevant to video data collection. Four additional variables were included in the video model: percent hardbottom (*ph*), maximum substrate relief (*rel*), current direction (*cur*), and water clarity (*wc*). Again, no predictor variables exhibited multicollinearity based on the variance inflation factors [33].

Our video GAM was coded as:

$$y = f(year) + s(depth) + s(temp) + s(doy) + s(pos) + s(ph) + f(rel) + f(cur) + f(wc), \quad (2)$$

where *y* is the video SumCount of red porgy, *year* is year of the sample, *depth* is bottom depth of the sample, *temp* is the bottom temperature, *doy* is day of the year, *pos* is the two-dimensional position variable, *ph* is the percent hardbottom, *rel* is the maximum substrate relief, *cur* is the current direction, *wc* is the water clarity, *f* is a categorical function, and *s* is a cubic spline (smoothed) function. The best fitting video SumCount model was again a Gaussian error distribution with a log-transformation of red porgy video SumCount. We compared the full model containing all predictor variables to reduced models that contained fewer predictor variables using ΔAIC.

We examined relationships between red porgy video SumCount and individual predictor variables that were included in the final (best) video model. We used the final video GAM to predict red porgy video SumCounts at average values of all other predictor variables. For predictions using categorical variables, we used the year 2015, a maximum substrate relief of 'moderate', and a current direction of 'away.' We also predicted red porgy video SumCount across the study area using the latitude, longitude, and depth of each cell and average values of all other covariates, as well as the categorical variable levels as described above.

## Recruitment failure hypothesis

The third objective of our work was to determine if recruitment failure was responsible for the decline in red porgy relative abundance (see *Results* below). We tested the recruitment failure hypothesis with two additional GAMs. The first tested whether the mean size of red porgy has increased over time, as would be expected if a fish species was experiencing recruitment failure. Alternatively, fish size would be expected to become smaller over time if fishing was primarily responsible for declines in relative abundance because most fisheries tend to have minimum size limits and therefore target larger individuals [37].

The length GAM related the mean length of red porgy caught in traps (1990–2021) to various predictor variables. Only traps that caught red porgy were included in the analysis. Here, we calculated the mean total length of all red porgy caught in each trap and related that to three predictor variables that we *a priori* hypothesized might influence red porgy mean length:

year, depth, and position. Red porgy caught by SERFS were measured for fork length or total length over time, so we converted fork lengths to total lengths using the equation provided by Potts and Manooch [38]. We weighted trap samples by the total number of red porgy caught in each trap, so that mean length based on many fish in a trap was weighted more heavily than a mean length from a single fish caught in the trap. The best fitting GAM included a Gaussian error distribution and no data transformation.

The last GAM addressed the recruitment failure hypothesis in a somewhat different way by specifically modeling the relative abundance of juvenile red porgy caught in chevron traps (1990–2021). Previous work has shown that the length at which red porgy reach 50% maturity is 290 mm total length [39], so we used that length cutoff to define juveniles here. This last GAM related the trap catch of juvenile red porgy to the same six predictor variables as were included in Eq 1 above. The best fitting juvenile GAM model used a Tweedie error distribution, likely due to increased zero inflation, and a log transformation. If recruitment failure was occurring, we hypothesized that the rate of decline of juvenile red porgy in the chevron trap survey would be greater than the decline observed for all red porgy caught in traps, e.g., [40].

## Results

Over the 32-year time span of chevron trap sampling, 20,920 trap samples were included in our analyses, ranging from a low of 202 chevron traps in 2003 to a high of 1,904 in 2021 (Table 1). A total of 13,081 videos were included in the analyses from 2011 to 2021, ranging from a low of 580 in 2011 to 1,654 in 2018. The depth distribution of samples was relatively consistent across the study, while the latitude range of sampling expanded somewhat over time (Table 1). A benefit of including depth and position in our GAMs is that they standardize for modest changes in the spatial distribution of sampling over time.

### Temporal and spatial patterns of red porgy

The observed proportion of trap and video samples having a positive trap catch or video observation of red porgy declined over the study (Fig 2). For traps, the proportion positive was around 0.5 in the early 1990s, but declined to approximately 0.2 by the late 2010s and to around 0.1 by 2021 (83% decline overall). Although a shorter time series, the proportion positive for red porgy on video also declined from around 0.4 in the early 2010s to just over 0.1 by 2021 (70% decline). The biggest change in proportion positive for red porgy occurred between 2019 and 2021, where declines in relative abundance were 49% in traps and 54% on video (Fig 2).

A total of 32,181 red porgy were caught in chevron traps in this study. Mean observed (unstandardized) trap catch and video SumCounts declined 89% and 73% respectively over the study (Fig 2). Mean trap catch (number of red porgy individuals caught per trap) was high and variable from the 1990s through the late 2000s, but has been lower and declining since the late 2000s and especially low in 2021. Mean video SumCount of red porgy has been variable in the 2010s, but was substantially lower in 2021 compared to earlier years (Fig 2).

Observed chevron trap catches of red porgy were higher in the northern part of the study area, especially in deeper waters, while catch rates were generally lower further south in Georgia and Florida (Fig 1A). Observed video SumCounts of red porgy displayed a similar distribution, being higher in the northern areas of the study area (i.e., North and South Carolina) and lower in Georgia and Florida, aside from some higher video counts in some cells off Cape Canaveral, Florida (Fig 1B).

The best fitting GAM for red porgy trap catch was the full model that included all six predictor variables and explained 26.9% of the deviance in trap catch (Table 2). Spatial position

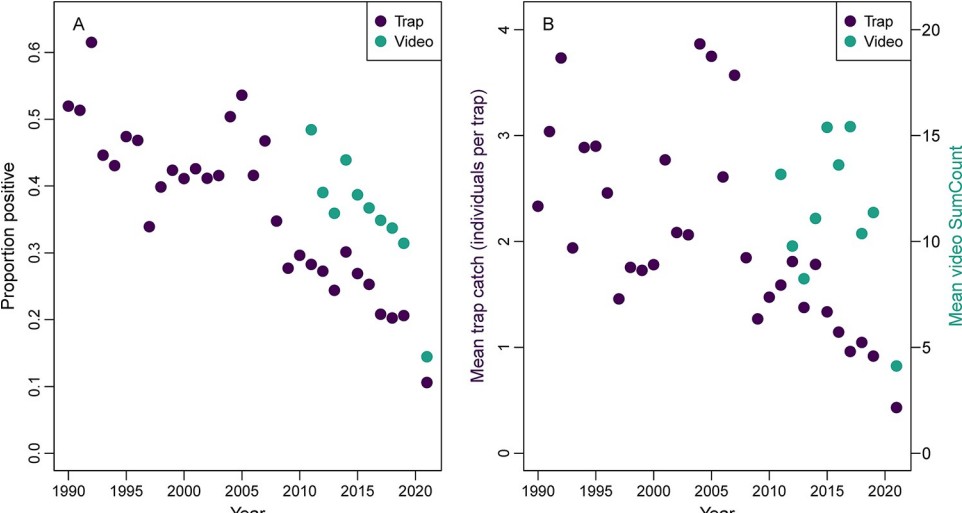

**Fig 2. Red porgy (*Pagrus pagrus*) percent occurrence, mean trap catch, and mean video counts from traps and video.** (A) Proportion of trap or video samples capturing or observing red porgy (*Pagrus pagrus*) from the Southeast Reef Fish Survey along the southeast United States Atlantic coast. (B) Mean trap catch (individuals per trap) or mean video SumCount of red porgy. Chevron trap data were collected from 1990 to 2021, while videos were collected from 2011 to 2021. Note no sampling occurred in 2020.

explained the most variability in trap catch, followed by depth and year; trap soak time, bottom water temperature, and especially day of the year explained very little deviance (Table 3). The second-best trap catch model that excluded bottom water temperature had a ΔAIC value of 6.8, suggesting much less support based on the data. The best fitting GAM for red porgy video SumCount included all predictor variables except water clarity, and this best model explained 35.9% of the deviance in video counts (Table 2). Similar to the trap catch model, position and depth explained the most variability in video SumCount, followed by current direction, year, and percent hardbottom; bottom water temperature, substrate relief, and day of the year explained very little of the variation in video counts (Table 3). The second-best video count model was the full model that had a ΔAIC value of 3.1, suggesting less support.

Standardized red porgy trap catches and video SumCounts declined over the study period, especially since the mid-2000s (Fig 3). In terms of standardized trap catches of red porgy, the last four years in the time series (2017–2021) were the lowest values over the 32-year time series. Red porgy declined by 77% between the year of highest mean red porgy relative abundance (1992) and the year of lowest mean red porgy relative abundance (2021), and 32% of that decline occurred between 2019 and 2021. Precision around annual relative abundance estimates was high, with a mean annual coefficient of variation of 0.08 (annual range = 0.06–0.11). Standardized video SumCounts of red porgy declined nearly linearly by 69% since video data collection began in 2011, and the largest decline occurred between 2019 and 2021 (45%). The precision of video-based relative abundance estimates was high (mean coefficient of variation = 0.08; range = 0.07–0.15). There was a strong correlation between standardized trap catch and video SumCounts (r = 0.80) during the 2011 to 2021 time period when both sets of data were collected, although video SumCounts appeared to decline slightly faster than trap catch (Fig 3).

There were strong spatial patterns in standardized red porgy trap catches and video Sum-Counts in the SEUS, as evidenced by the spatial position predictor variable being the most important variable in both trap and video GAMs (Table 3, Fig 4). The spatial predictions from

**Table 2. Model selection for the spatially explicit generalized additive models of trap catch of all red porgy (*Pagrus pagrus*; 1990–2021), counts of all red porgy on video (2011–2021), mean length of red porgy (1990–2021), or trap catch of juvenile red porgy only (1990–2021) from data collected by the Southeast Reef Fish Survey along the southeast United States Atlantic coast.**

| Model | ΔAIC | DE | f(year) | s(depth) | s(temp) | s(doy) | s(dur) | s(pos) | f(cur) | f(wc) | f(rel) | s(ph) |
|---|---|---|---|---|---|---|---|---|---|---|---|---|
| **Trap—all fish** | | | | | | | | | | | | |
| Full | 0.0 | 26.9 | 30*** | 2.0*** | 8.6* | 8.3*** | 1.8* | 28.9*** | na | na | na | na |
| Full−*temp* | 6.8 | 26.8 | 30*** | 2.0*** | ex | 8.3*** | 1.8* | 28.9*** | na | na | na | na |
| Full−*dur* | 7.0 | 26.9 | 30*** | 2.0*** | 8.6* | 8.3*** | ex | 28.9*** | na | na | na | na |
| Full−*temp*−*dur* | 13.8 | 26.8 | 30*** | 2.0*** | ex | 8.3*** | ex | 28.9*** | na | na | na | na |
| **Video—all fish** | | | | | | | | | | | | |
| Full−*wc* | 0.0 | 35.9 | 9*** | 2.0*** | 3.9*** | 1.0*** | na | 28.9*** | 2*** | ex | 2*** | 7.2*** |
| Full | 3.1 | 35.9 | 9*** | 2.0*** | 3.8*** | 1.0*** | na | 28.9*** | 2*** | 2 | 2*** | 7.2*** |
| Full−*wc*−*temp* | 19.6 | 35.8 | 9*** | 2.0*** | ex | 2.9*** | na | 28.9*** | 2*** | ex | 2*** | 7.2*** |
| Full−*temp* | 23.1 | 35.8 | 9*** | 2.0*** | ex | 3.0*** | na | 28.9*** | 2*** | 2 | 2*** | 7.2*** |
| **Mean length** | | | | | | | | | | | | |
| Full | 0.0 | 45.0 | 30*** | 2.0*** | na | na | na | 28.7*** | na | na | na | na |
| Full−*depth* | 54.4 | 44.5 | 30*** | ex | na | na | na | 28.7*** | na | na | na | na |
| Full−*pos* | 930.2 | 35.2 | 30*** | 2.0*** | na | na | na | ex | na | na | na | na |
| Full−*depth*−*pos* | 1185.2 | 32.3 | 30*** | ex | na | na | na | ex | na | na | na | na |
| **Trap—juveniles only** | | | | | | | | | | | | |
| Full−*dur*−*doy* | 0.0 | 44.8 | 30*** | 8.7*** | 1.0* | ex | ex | 28.2*** | na | na | na | na |
| Full−*dur* | 0.7 | 44.8 | 30*** | 8.7*** | 1.0 | 1.0 | ex | 28.2*** | na | na | na | na |
| Full−*doy* | 2.5 | 44.8 | 30*** | 8.7*** | 1.0* | ex | 1.9 | 28.2*** | na | na | na | na |
| Full | 3.0 | 44.8 | 30*** | 8.7*** | 1.0 | 1.0 | 2.0 | 28.2*** | na | na | na | na |

Full models include all model covariates, while the minus sign followed by a covariate name indicates that covariate was excluded from the full model. Degrees of freedom are shown for factor (*f*) terms and estimated degrees of freedom are shown for smoothed terms (*s*). Asterisks denote significance at the following alpha levels: *0.05, **0.01, ***0.001; ΔAIC = delta Akaike information criterion (best model ΔAIC = 0.0); DE = percent deviance explained by the model; *y* = year of the sample; *depth* = bottom depth; *temp* = bottom water temperature; *doy* = day of the year; *dur* = trap soak time; *pos* = position of the sample; *cur* = current direction relative to the video camera; *wc* = water clarity; *rel* = maximum substrate relief; *ph* = percent of visible bottom substrate that was hardbottom; ex = covariate was excluded from that particular model based on ΔAIC; and na = covariate was not applicable to that particular model.

trap and video models were very similar, both indicating high trap catches and video counts in the deeper areas of southern North Carolina and South Carolina. Trap catches and video Sum-Counts were low in northern North Carolina, Florida, and inshore areas across the entire study area. The biggest difference between trap and video spatial predictions was the slightly higher predictions in some inshore areas from the trap GAM compared to the video GAM (Fig 4).

**Table 3. Deviance explained by each model covariate in generalized additive models built on data collected by the Southeast Reef Fish Survey along the southeast United States Atlantic coast.**

| Model | Best model | *year* | *depth* | *temp* | *doy* | *dur* | *pos* | *cur* | *wc* | *rel* | *ph* |
|---|---|---|---|---|---|---|---|---|---|---|---|
| Trap–all fish | 26.9 | 6.0 | 8.3 | 1.0 | 0.4 | 1.2 | 23.0 | - | - | - | - |
| Video–all fish | 35.9 | 2.1 | 11.1 | 1.3 | 0.7 | - | 30.9 | 2.3 | - | 1.0 | 1.9 |
| Mean length | 45.0 | 32.4 | 19.3 | - | - | - | 26.7 | - | - | - | - |
| Trap–juveniles only | 44.8 | 19.6 | 11.8 | 1.1 | - | - | 28.9 | - | - | - | - |

Deviance explained by each covariate was calculated from models that only included each single covariate by itself, and the deviance explained for the best model was taken from final models included in Table 2. Empty cells imply that covariate was not included in the best model, so no deviance explained is provided.

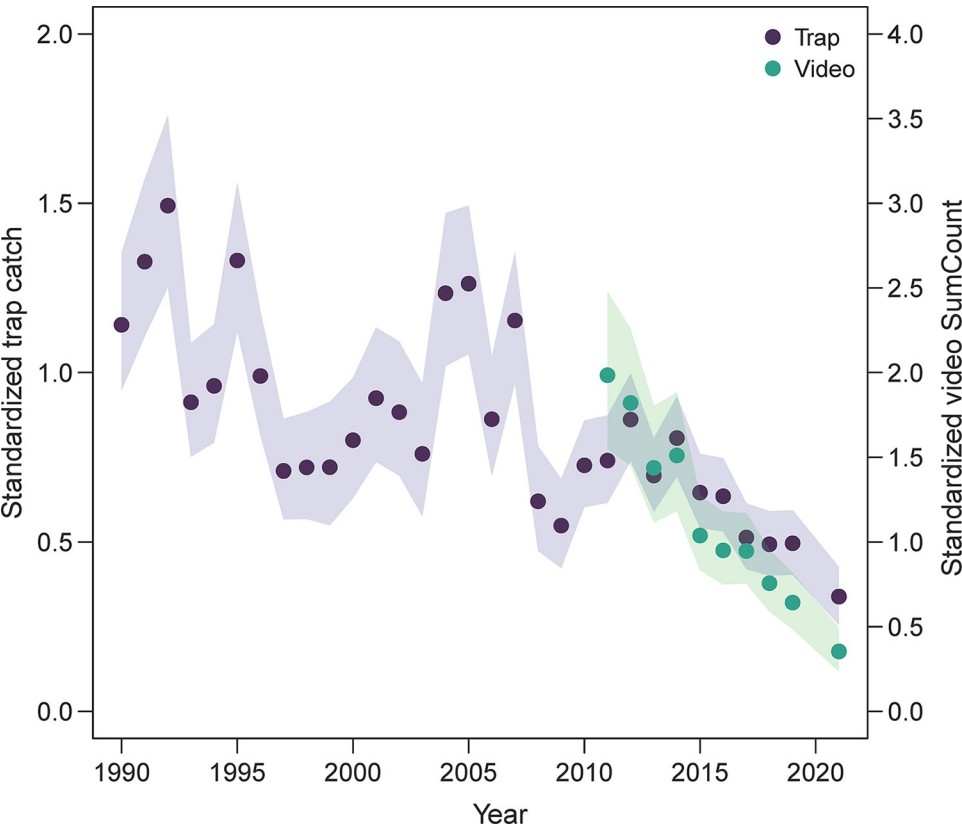

**Fig 3. Standardized trap catches and video counts of red porgy (*Pagrus pagrus*) using spatially explicit generalized additive models from the Southeast Reef Fish Survey along the southeast United States Atlantic coast.** Points indicate mean values (traps = blue; video = green) and shaded areas indicate 95% confidence intervals.

Depth explained much more variability in red porgy trap catch than day of the year, bottom temperature, or trap soak time (Table 3, Fig 5). There was a strong positive relationship between red porgy trap catch and depth, with the lowest standardized trap catches found less than 50 m deep and the highest standardized trap catches found in the deepest water sampled in our study (i.e., 80–100 m deep). Generally, red porgy trap catches declined throughout the summer, were weakly related to bottom water temperature, and were slightly positively related to trap soak time (Fig 5).

Depth was also strongly positively related to red porgy video SumCounts, while the remaining video covariates were of lesser importance (Table 3, Fig 6). Similar to the depth effect in trap GAM, red porgy video SumCounts were low at 15–60 m deep but increased substantially and positively from 60 to 100 m deep. There was a weak negative relationship between red porgy video SumCount and day of the year and a weak positive relationship with bottom water temperature. Although red porgy video SumCounts were positively related to percent hardbottom, they were negatively related to maximum substrate relief. Last, red porgy were more likely to be counted on video when the camera was facing down-current and less likely to be counted when the camera was facing up-current.

## Recruitment failure hypothesis

Observed mean lengths of red porgy were variable over space but were generally higher in deeper waters and lower inshore or in mid-continental shelf waters between North Carolina

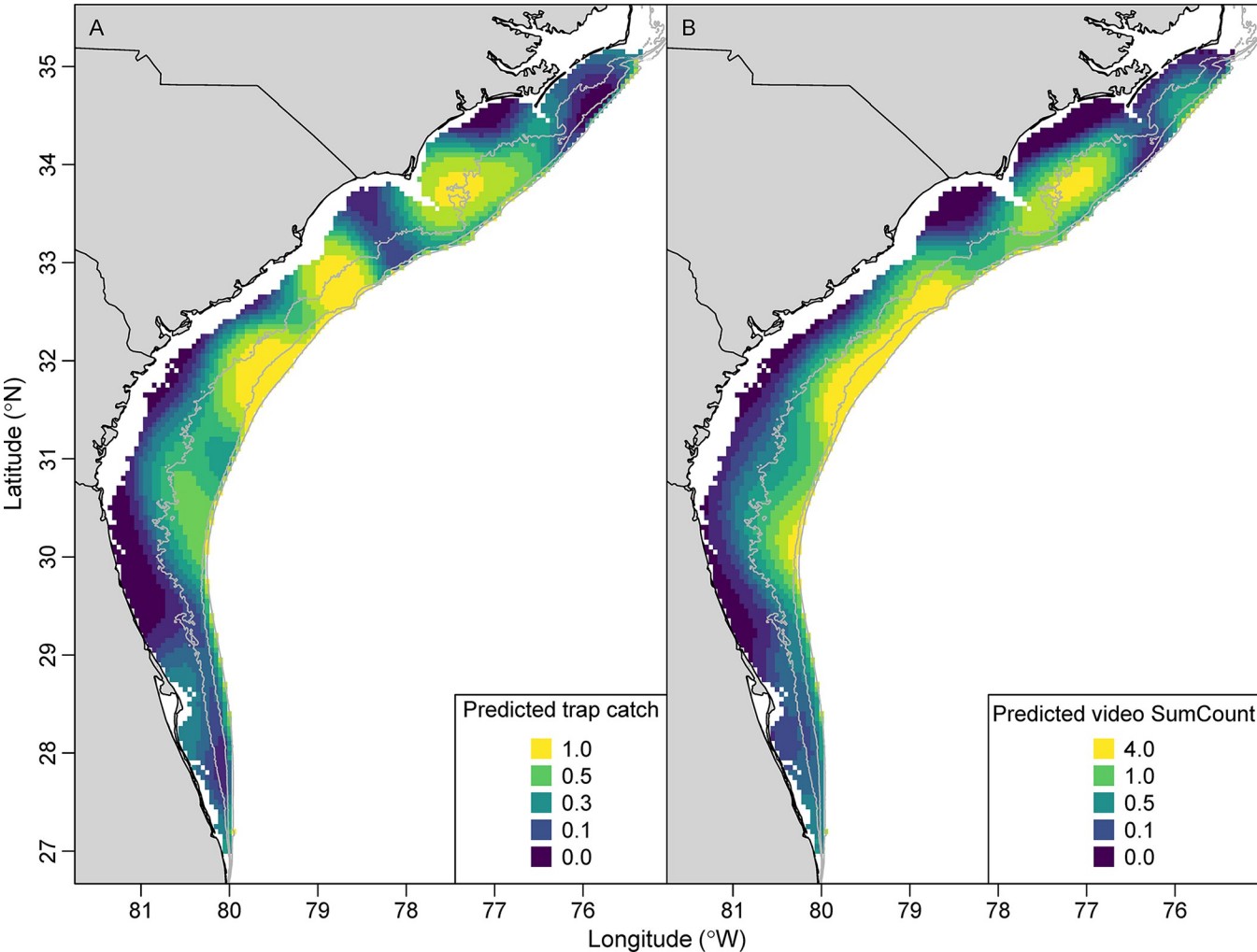

**Fig 4.** Predicted trap catch of the number of individuals (A; 1990–2021) and video SumCounts (B; 2011–2021) of red porgy (*Pagrus pagrus*) from the Southeast Reef Fish Survey along the southeast United States Atlantic coast. Predictions were based on the spatial position and depth of each cell at average values of all other model covariates using spatially explicit generalized additive models. Gray isobaths are 30, 50, and 100 m.

and Florida (Fig 1C). Most juvenile red porgy (< 290 mm total length) were caught in mid-continental shelf waters of North and South Carolina, but were highly patchy and variable across space (Fig 1D).

The full GAM relating the mean length of red porgy to three predictor variables was selected over reduced models and explained 45.0% of the model deviance (Table 2). Each of these three predictor variables explained a similar amount of model deviance, with year explaining the most and depth explaining the least (Table 3). The second-best model excluded depth but had a ΔAIC of 54.4, suggesting no support for this model (Table 2). The best GAM describing the catch of juvenile red porgy included four predictor variables: year, depth, bottom temperature, and spatial position; day of the year and trap soak time were excluded (Table 2). This model explained 44.8% of the model deviance but was only 0.7 AIC units better than the second-best model that only excluded trap soak time. For the best juvenile red porgy GAM, spatial position explained the most deviance, followed by year, depth, and bottom water temperature (Table 3).

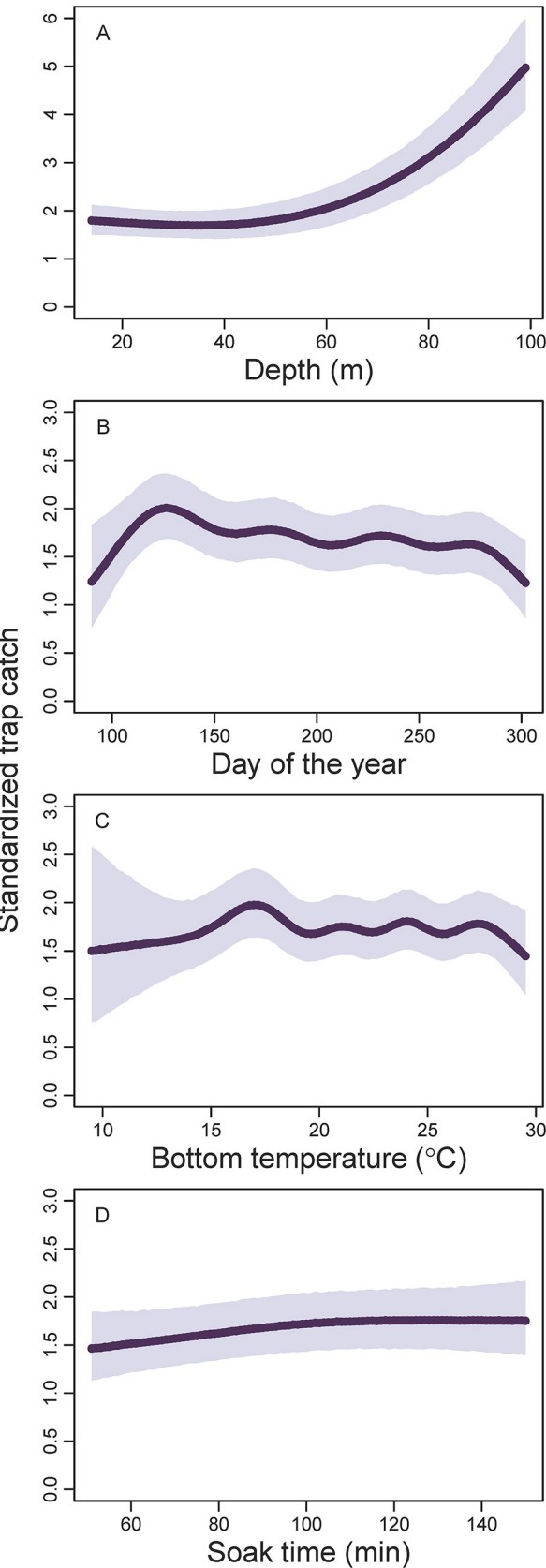

**Fig 5.** Standardized trap catch of red porgy (*Pagrus pagrus*) as a function of (A) depth (m), (B) day of the year, (C) bottom temperature (˚C), or (D) soak time (min) using spatially explicit generalized additive models from data collected by the Southeast Reef Fish Survey along the southeast United States Atlantic coast, 1990–2021. Thick black lines are mean values at average values of all model covariates and shaded areas are 95% confidence intervals. Note the varying *y*-axis range among panels.

Standardized mean length of red porgy generally increased over the 32-year study (Fig 7). Standardized mean length was approximately 300 mm total length in the early 1990s, but increased to ~340 mm total length by the 2000s and ~360 mm total length by the 2010s, an overall increase of 29%. Standardized mean total length of red porgy was shortest in shallow water (< 30 m deep) and highest in 60–80 m deep (Fig 7). Standardized mean length was highest in deeper waters across the study area, but also inshore in North Carolina and northern South Carolina (Fig 8). Standardized mean length was lowest in northern North Carolina and inshore in Florida and Georgia.

Standardized trap catches of juvenile red porgy were high but variable in the early 1990s and lower through the 2000s, but catches of juvenile red porgy mostly disappeared from the

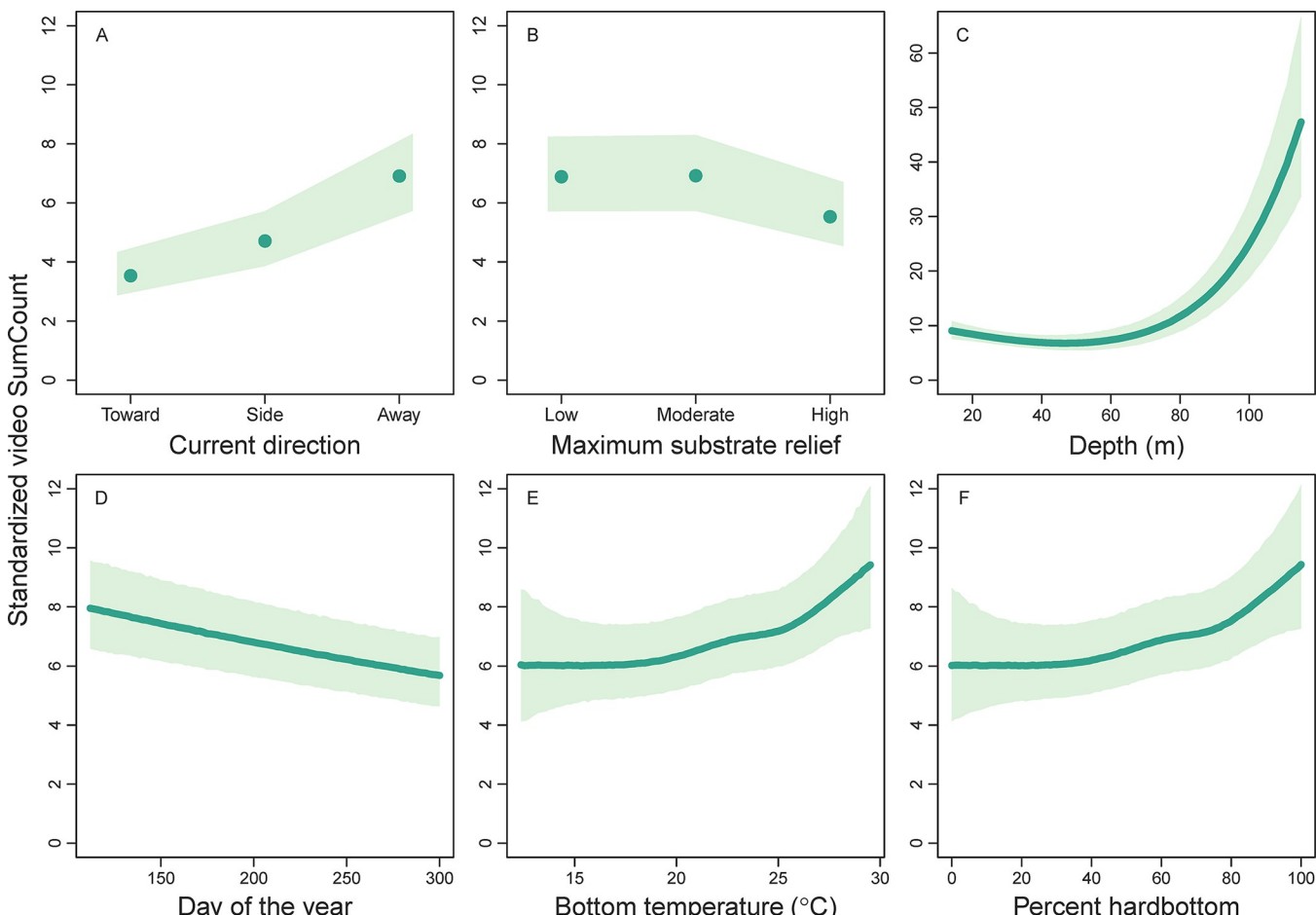

**Fig 6.** Standardized video SumCounts of red porgy (*Pagrus pagrus*) as a function of (A) current direction, (B) maximum substrate relief, (C) depth (m), (D) day of the year, (E) bottom temperature (˚C), and (F) percent hardbottom using spatially explicit generalized additive models from data collected by the Southeast Reef Fish Survey along the southeast United States Atlantic coast, 2011–2021. Red points or lines are mean values at average values of all model covariates and shaded areas are 95% confidence intervals. Note the varying *y*-axis range among panels.

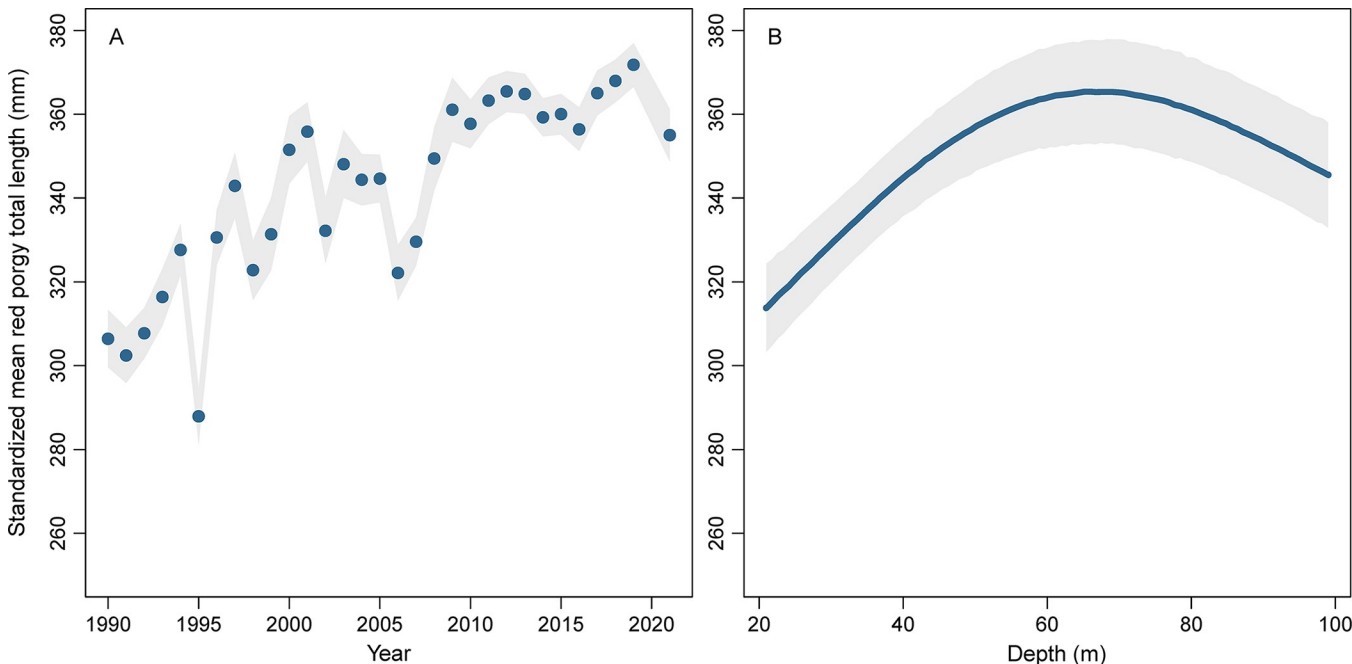

**Fig 7.** Standardized total length (mm) of red porgy (*Pagrus pagrus*) caught in chevron traps as a function of (A) year or (B) bottom depth (m) using spatially explicit generalized additive models from data collected by the Southeast Reef Fish Survey along the southeast United States Atlantic coast, 1990–2021. Black points or lines are mean values at average values of all model covariates and shaded areas are 95% confidence intervals.

late 2000s through 2021 (Fig 9). Mean trap catch of juvenile red porgy declined by 99% between the highest (1995) and lowest years (2018). Juvenile red porgy were mostly caught between 30 and 40 m deep and again around 100 m deep, but note that the precision of those deeper estimates was very low (Fig 9). There was also a weak relationship between the standardized trap catch of juvenile red porgy and bottom water temperature (Fig 9). The spatial distribution of juvenile red porgy from the juvenile red porgy GAM was highest in mid- or outer continental shelf habitats from southern North Carolina to Georgia and lowest inshore, especially in Georgia and Florida (Fig 8).

## Discussion

Using trap catches and video counts, we found that red porgy relative abundance has declined severely over the last 30 years in the SEUS. While these declines are substantial, the decline in trap catches of juvenile red porgy was much greater and mean length of red porgy increased over the same time period, suggestive of recruitment failure occurring in red porgy over the last two decades. Both chevron traps and underwater video sampled red porgy relative abundance well in the SEUS, as evidenced by high precision of estimates, appropriate model diagnostics, strong correspondence between observed and predicted data, and consistency of results between gears. Red porgy abundance has continued to decline in recent years despite strict bag and size regulations and relatively low commercial and recreational landings. Taken together, these results suggest that current minimum size and bag limits may be inadequate to sustainably manage red porgy in the SEUS given that external forces appear to be limiting recruitment in the region.

The decline of red porgy relative abundance estimated over our study period (69–77%) was similar to previous findings. Vaughan and Prager [16] estimated an 89% reduction in

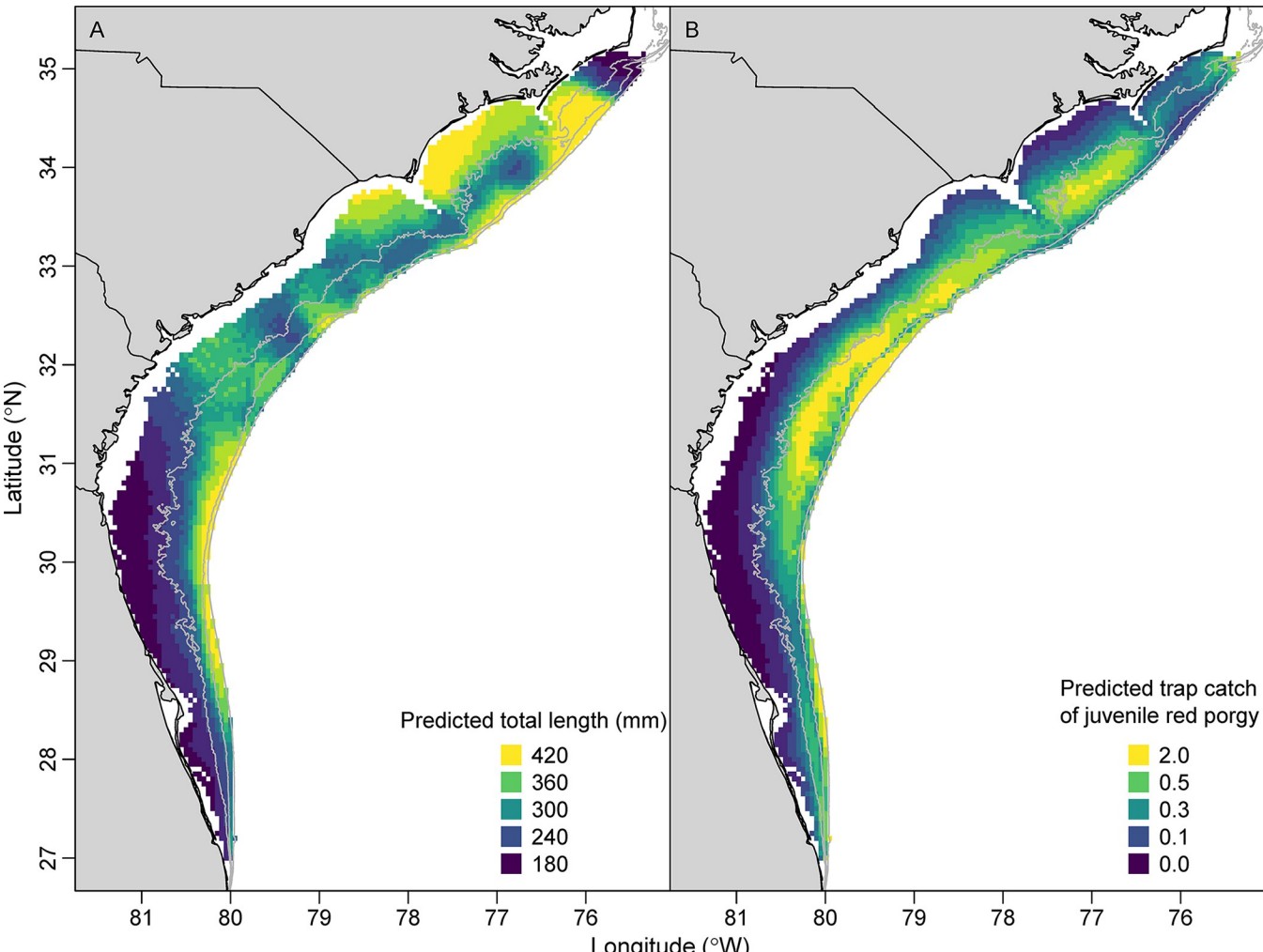

**Fig 8. Spatial predictions of total length (mm) and trap catch of juvenile (< 290 mm) red porgy (*Pagrus pagrus*) from the Southeast Reef Fish Survey along the southeast United States Atlantic coast, 1990–2021.** Predictions were based on the spatial position and depth of each cell at average values of all other model covariates using spatially explicit generalized additive models. Gray isobaths are 30, 50, and 100 m.

spawning stock biomass of red porgy in the SEUS from 3,570 t in 1979 to 397 t in 1997, suggesting that red porgy relative abundance had already declined substantially before chevron trapping for the present study began in 1990. Analyzing a subset of chevron trap data from the present study, Smart et al. [18] used the Vector-Autoregressive Spatio-Temporal model [41,42] to show that adult and recruit red porgy relative abundance was low in the most recent years of the time series compared to the late 1990s. The most recent red porgy stock assessment in the SEUS estimated declines in red porgy abundance by about 75% between the late 1970s and the early 2000s, with additional declines during the 2010s [17]. Multiple sources of information indicate red porgy have declined substantially since the 1990s, but severe declines in abundance also preceded the time period of our study [15–17], suggesting that the decline of red porgy relative abundance observed here is underestimated.

Recruitment failure appears to be at least partly responsible for the decline of red porgy abundance in the SEUS. We documented a 99% decline in juvenile red porgy caught in chevron traps over time, which is much greater than the decline in trap catch rates for all sizes of red porgy in our study and similar to previous estimates [16–18,43]. We also documented a

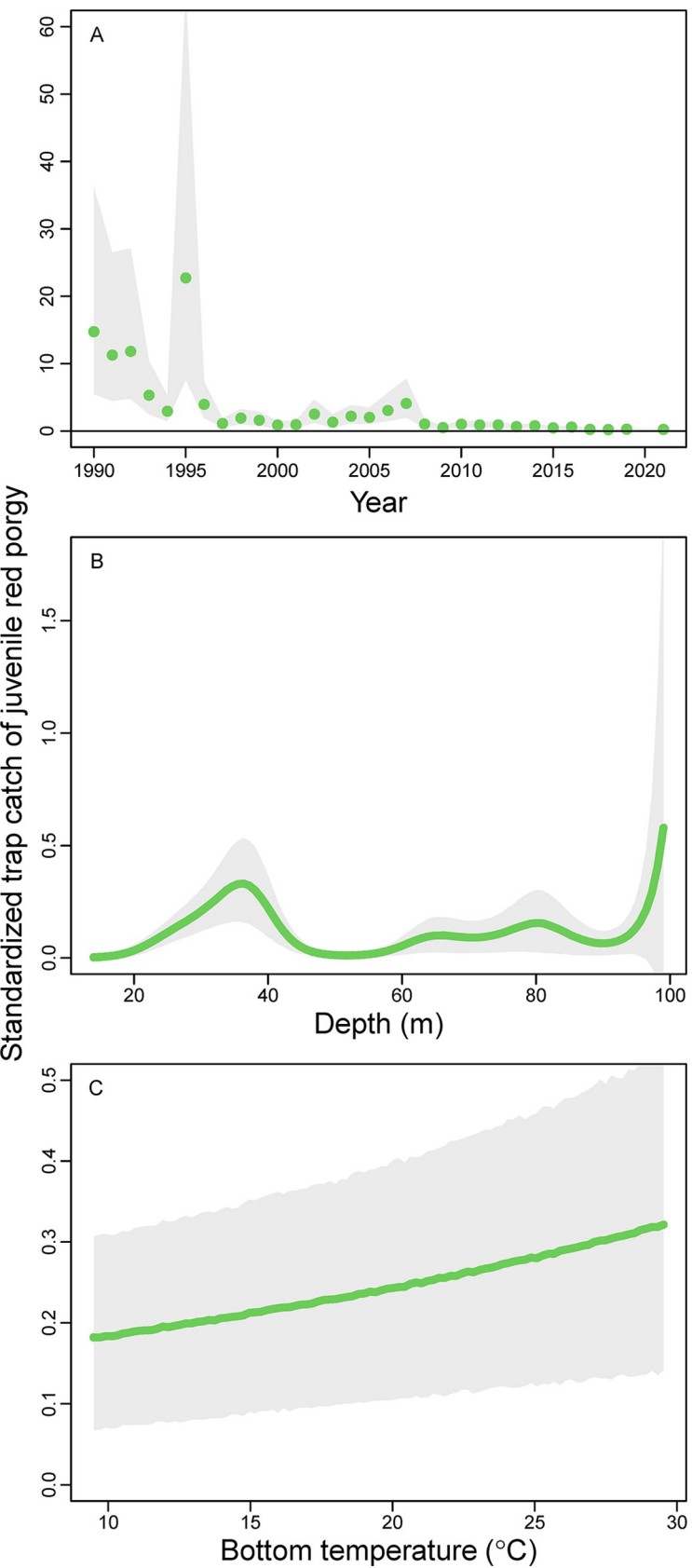

**Fig 9.** Standardized trap catch of juvenile red porgy (*Pagrus pagrus*; < 290 mm) as a function of (A) year, (B) depth, and (C) bottom temperature (˚C) using spatially explicit generalized additive models from data collected by the Southeast Reef Fish Survey along the southeast United States Atlantic coast, 1990–2021. Solid points or thick black lines are mean values at average values of all model covariates and shaded areas are 95% confidence intervals. Note the varying *y*-axis range among panels.

29% increase in mean size of red porgy over time, which is consistent with a recruitment failure hypothesis and the opposite of what would be expected from severe overfishing, where mean size generally declines [37,44,45]. It is possible that some of the increase in mean size of red porgy was due to increased individual growth rates due to density dependence at low abundance, although direct evidence for increased growth rates in recent years is lacking. Low recruitment is not just limited to red porgy–some other economically important fish species in the SEUS have also displayed lower than expected recruitment in recent years including red grouper (*Epinephelus morio*), scamp (*Mycteroperca phenax*), black sea bass (*Centropristis striata*), and gag (*Mycteroperca microlepis*) [40,43].

The mechanism for why recruitment is limited for these species has not yet been identified, but there are two main possibilities. The first is recruitment overfishing, where fishing reduces spawning biomass to a level where reproductive output is limited due to Allee effects or egg or sperm limitation from skewed sex ratios [46,47]. A second possibility is that the mortality of eggs, larvae, or juveniles has increased. Some potential reasons mortality may have increased on young fish include changes in winter-time environmental conditions [48,49], increased predation by species such as lionfish (*Pterois volitans*) [50] or red snapper (*Lutjanus campechanus*) [51], or the disappearance of food resources that support larval or juvenile red porgy. There are currently no regional-scale surveys in the SEUS that consistently collect eggs, larvae, or small juvenile reef fishes such as red porgy, making it difficult to determine the mechanism for low recruitment. Identifying why low recruitment is occurring for red porgy and other economically important species in the SEUS is critically important.

Despite already being at historically low levels of abundance, red porgy declined dramatically between 2019 and 2021–32% in traps and 45% on video. The reason for this unusually large decline in red porgy abundance is unclear but may be related to the COVID-19 pandemic. Recreational fishing effort appeared to increase in many places during the first year of the COVID-19 pandemic [52,53], in part due to many first-time anglers [54]. In a tributary of Lake Huron, for instance, angler exploitation rate of rainbow trout (*Oncorhynchus mykiss*) decreased by half during lockdowns in early 2020 compared to before the pandemic, but an eight-fold increase in exploitation quickly followed once travel restrictions were eased in the fall of 2020 [55]. We are not aware of fishing effort trends in the SEUS during the COVID-19 pandemic, but it is possible that increased fishing effort for reef-associated fish species during 2020 and early 2021 contributed to further declines of red porgy either directly through retention or bycatch mortality [56,57]. But the disappearance of juvenile red porgy cannot likewise be explained by an increase in fishing effort due to the COVID-19 pandemic because a 356-mm minimum size limit exists for red porgy in the region.

Red porgy adults and juveniles were not found homogenously throughout the SEUS. Instead, adult red porgy were mostly encountered in deeper, outer continental shelf or shelf break waters between southern North Carolina and Georgia, consistent with the findings of previous studies [18,22,58]. In our study, there was a high degree of consistency between the spatial predictions from trap and video GAMs, suggesting the predictions were tracking the true spatial distribution of red porgy in the SEUS. Although red porgy were encountered across a wide range of depths in our study, they strongly preferred deeper compared to shallower water. Our results make sense in light of Manooch and Hassler [14] stating that red

porgy were commonly found at depths from 18 to 183 m throughout their range, with the deepest individual collected in 280 m of water south of the Canary Islands [59]. In contrast, Wheeler [60] suggested that red porgy were caught in depths of only up to 60 m in the Mediterranean Sea and Atlantic Ocean off north Africa.

There is a paucity of information on the spatial and depth distributions of juvenile red porgy in the literature. We found juvenile red porgy in mid- to outer-continental shelf habitats in 35–40 m of water between southern North Carolina and Georgia in our study, generally inshore of adult distributions. Manooch [61] indicated that it was probable that young red porgy were distributed inshore of adult populations, and he reported a young-of-the-year red porgy captured in a trawl in 9 m of water off Charleston, South Carolina. Wheeler [60] noted that young red porgy can also be caught near shore in the Mediterranean Sea. The discrepancy in depth distributions among these studies may be partially explained by the specific age of the sampled red porgy; age-0 and age-1 red porgy may occur in shallow water, as indicated by Wheeler [60] and Manooch [61], but older juvenile red porgy may move into somewhat deeper continental shelf waters by the time they become selected for by chevron traps (age 2 or 3) [17]. But note that Smart et al. [18] predicted that most red porgy recruits occur in outer shelf waters between North Carolina and Georgia, somewhat deeper than we found in our study. The relative scarcity of early juvenile red porgy in the SEUS is similar to other species in the region like red snapper [62].

Red porgy are reef generalists and occur over a wide range of reef habitats in the SEUS including rocky ledges, flat limestone pavement, and sandy habitats adjacent to reefs [14,22]. Yet red porgy did not use all seafloor habitats equally in our study. Instead, red porgy mainly occupied habitats with high percent hardbottom and low substrate relief, suggesting that red porgy tend to prefer continuous (i.e., pavement) habitats that lack vertical structure.

There were some drawbacks of our study. First, observational studies employing regression techniques like ours are correlational in nature and therefore causation cannot be determined or implied. Second, our sampling footprint expanded somewhat over time, particularly to the south in 1997 and to the north in 2012. To avoid potential concerns with survey expansion over time, we used a GAM that standardized for changes in the spatial footprint of the survey so that our estimates of red porgy abundance were not confounded with changes in the sampling distribution. Third, recruitment was so limited in recent years that the number of juvenile red porgy caught in chevron traps was small, which resulted in higher uncertainty around GAM predictions. Fourth, we used the mean length of red porgy caught in chevron traps in our length GAM, but using all fish lengths may be more informative; we elected for simplicity over complexity. Last, our models explained 27–45% of the deviance among models, meaning a majority of the model deviance for trap catch, video counts, or mean length remained unexplained. The spatial or temporal distribution of prey and predators of red porgy, social interactions, or other unmeasured environmental or habitat variables might be responsible for the remaining unexplained deviance.

Our study indicates that red porgy has experienced severe declines in abundance over the last three decades, and given that significant declines occurred previous to our study [16], the abundance of red porgy in the SEUS is likely at unprecedented low levels. The most recent stock assessment for red porgy indicated that spawning stock biomass of red porgy in 2017 was approximately 20% of the level corresponding to maximum sustainable yield [17], and our results suggest substantial declines of red porgy since that time between 2017 and 2021 (trap GAM = 32% decline; video GAM = 45% decline). Given that recruitment failure of red porgy is at least partially responsible for abundance declines, even the most drastic of management measures such as a prohibition of harvest is unlikely to result in significant increases in abundance until the low recruitment issue can be identified and remedied.

## Acknowledgments

We thank past and present SERFS staff members, numerous volunteers, and the captains and crews of the NOAA Ship *Nancy Foster*, NOAA Ship *Pisces*, R/V *Palmetto*, R/V *Savannah*, and SRVx *Sand Tiger* for data collection. Mention of trade names or commercial companies is for identification purposes only and does not imply endorsement by the National Marine Fisheries Service, NOAA. The scientific results and conclusions, as well as any views and opinions expressed herein, are those of the authors and do not necessarily reflect those of any government agency.

## Author Contributions

**Conceptualization:** Nathan M. Bacheler, Nikolai Klibansky.

**Data curation:** Nathan M. Bacheler.

**Formal analysis:** Nathan M. Bacheler, Nikolai Klibansky, Walter J. Bubley, Tracey I. Smart.

**Funding acquisition:** Nathan M. Bacheler, Walter J. Bubley, Tracey I. Smart.

**Investigation:** Nathan M. Bacheler.

**Methodology:** Nathan M. Bacheler, Nikolai Klibansky, Walter J. Bubley.

**Project administration:** Nathan M. Bacheler.

**Writing – original draft:** Nathan M. Bacheler.

**Writing – review & editing:** Nathan M. Bacheler, Nikolai Klibansky, Walter J. Bubley, Tracey I. Smart.

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
