## [Decision Letter · Decision Letter 0]

3 May 2023

PONE-D-23-04623Low recruitment drives the decline of red porgy (Pagrus pagrus) along the southeast USA Atlantic coast: inferences from fishery-independent trap and video monitoringPLOS ONE

Dear Dr. Bacheler,

Thank you for submitting your manuscript to PLOS ONE. After careful consideration, we feel that it has merit but does not fully meet PLOS ONE’s publication criteria as it currently stands. Therefore, we invite you to submit a revised version of the manuscript that addresses the points raised during the review process.

The reviewer suggested that there is an error (on page 18, line 373) - the wrong figure is referred to - This does indeed appear to be an error which should be changed. The other comments from the reviewer should be viewed as suggestions that the authors are free to ignore as these are not material to whether the manuscript meets PLOS ONE's publication criteria.

We look forward to receiving your revised manuscript.

Kind regards,

John A. B. Claydon, Ph.D.

Academic Editor

PLOS ONE

“The survey is funded by the U.S. National Marine Fisheries Service, but the authors received no specific funding for the analyses.”

Please respond by return e-mail so that we can amend your financial disclosure and competing interests on your behalf.

3. We note that Figures 1, 4 and 8 in your submission contain [map/satellite] images which may be copyrighted. All PLOS content is published under the Creative Commons Attribution License (CC BY 4.0), which means that the manuscript, images, and Supporting Information files will be freely available online, and any third party is permitted to access, download, copy, distribute, and use these materials in any way, even commercially, with proper attribution. For these reasons, we cannot publish previously copyrighted maps or satellite images created using proprietary data, such as Google software (Google Maps, Street View, and Earth). For more information, see our copyright guidelines: http://journals.plos.org/plosone/s/licenses-and-copyright.

a. You may seek permission from the original copyright holder of Figures 1, 4 and 8 to publish the content specifically under the CC BY 4.0 license. 

Additional Editor Comments:

The reviewer suggested that there is an error (on page 18, line 373) - the wrong figure is referred to - This does indeed appear to be an error which should be changed. The other comments from the reviewer should be viewed as suggestions that the authors are free to ignore as these are not material to whether the manuscript meets PLOS ONE's publication criteria.

My evaluation of the manuscript:

The authors use a large long-term (30-year) data set to explore a number of questions regarding red porgy that are pertinent to the fishery and species in general. The methods they use are appropriate, their analyses are statistically rigorous, and their conclusions are supported by their results. The manuscript is well written and (once the error identified by the reviewer has been addressed) appears to meet all PLOS ONE criteria for publication.

Reviewers' comments:

Reviewer's Responses to Questions

**Comments to the Author**

1. Is the manuscript technically sound, and do the data support the conclusions?

Reviewer #1: Yes

2. Has the statistical analysis been performed appropriately and rigorously? 

Reviewer #1: Yes

3. Have the authors made all data underlying the findings in their manuscript fully available?

Reviewer #1: Yes

4. Is the manuscript presented in an intelligible fashion and written in standard English?

Reviewer #1: Yes

5. Review Comments to the Author

Reviewer #1: An interesting paper that investigated the ways in which red porgy relative abundance and mean size varied in the southeast United States Atlantic coast. Based on the increase in the average length of individuals and the decline of juveniles over the time series, the authors conclude that the red porgy is currently experiencing low recruitment, which would be one of the reasons for the decline in species abundance.

The manuscript has a considerable amount of data (20,920 trap samples; 13,081 videos) over a 32-year time series. The data is well analyzed and interpreted and would make a solid contribution to fishing science, so I suggest that the manuscript be accepted for publication.

Throughout the text I found only a small mistake, on page 18, line 373, it mentions figure 1, but I believe that the sentence is about figure 2.

Below I suggest two small questions that are up to the authors to incorporate into the discussion or not, if they believe it is necessary.

In my opinion, a biological issue that I think might be important in debating recruitment failure was not raised in the discussion. What are the possible effects in relation to depletion that could be intensified with protogynous hermaphroditism, in terms of sex ratio and change in average size. With larger individuals currently found, wouldn't a higher proportion of males be expected in this population? I believe that the authors should raise this information in the literature and elucidate these speculations in the discussion about recruitment.

Another issue that caught my attention was that observed catches were higher in the northern part of the study area and in deeper waters. Is there the possibility of a stock shift to colder and deeper regions in response to environmental changes? Is there information about the species above North Carolina that could support this discussion?

6. PLOS authors have the option to publish the peer review history of their article (what does this mean?). If published, this will include your full peer review and any attached files.

Reviewer #1: No

---

## [Author Response · Author response to Decision Letter 0]

5 May 2023

Additional Editor Comments:

The reviewer suggested that there is an error (on page 18, line 373) - the wrong figure is referred to - This does indeed appear to be an error which should be changed. The other comments from the reviewer should be viewed as suggestions that the authors are free to ignore as these are not material to whether the manuscript meets PLOS ONE's publication criteria.

Thank you so much for catching this error – we have now changed the text to read “Fig. 2” on line 373. 

My evaluation of the manuscript:

The authors use a large long-term (30-year) data set to explore a number of questions regarding red porgy that are pertinent to the fishery and species in general. The methods they use are appropriate, their analyses are statistically rigorous, and their conclusions are supported by their results. The manuscript is well written and (once the error identified by the reviewer has been addressed) appears to meet all PLOS ONE criteria for publication.

Thank you very much for your review and editorial work on our manuscript. 

Reviewer #1: 

An interesting paper that investigated the ways in which red porgy relative abundance and mean size varied in the southeast United States Atlantic coast. Based on the increase in the average length of individuals and the decline of juveniles over the time series, the authors conclude that the red porgy is currently experiencing low recruitment, which would be one of the reasons for the decline in species abundance. The manuscript has a considerable amount of data (20,920 trap samples; 13,081 videos) over a 32-year time series. The data is well analyzed and interpreted and would make a solid contribution to fishing science, so I suggest that the manuscript be accepted for publication.

We thank Reviewer #1 for their thoughtful review of our manuscript. 

Throughout the text I found only a small mistake, on page 18, line 373, it mentions figure 1, but I believe that the sentence is about figure 2.

Thank you so much for catching this error – we have now changed the text to read “Fig. 2” on line 373.

Below I suggest two small questions that are up to the authors to incorporate into the discussion or not, if they believe it is necessary. In my opinion, a biological issue that I think might be important in debating recruitment failure was not raised in the discussion. What are the possible effects in relation to depletion that could be intensified with protogynous hermaphroditism, in terms of sex ratio and change in average size. With larger individuals currently found, wouldn't a higher proportion of males be expected in this population? I believe that the authors should raise this information in the literature and elucidate these speculations in the discussion about recruitment.

We had mentioned on lines 625-627 that recruitment overfishing could be occurring in red porgy due to sperm limitation from skewed sex ratios, but the reviewer rightly suggests that males could actually become more common than females as average sizes become larger. Thus, we now include “or egg” in the following sentence on line 627: “…due to Allee effects or egg or sperm limitation from skewed sex ratios [46,47].”

Another issue that caught my attention was that observed catches were higher in the northern part of the study area and in deeper waters. Is there the possibility of a stock shift to colder and deeper regions in response to environmental changes? Is there information about the species above North Carolina that could support this discussion?

In our manuscript, we quantified the current spatial distribution of red porgy in the region. However, it is beyond the scope of this manuscript to describe how red porgy might shift their spatial distribution in the face of future environmental changes in the region, which was not a part of our analyses. Based on analyses from the NEFSC trawl survey, which samples in the northeast USA down to central North Carolina, red porgy are only very rarely caught north of North Carolina.

---

## [Editor Report · Decision Letter 1]

9 May 2023

Low recruitment drives the decline of red porgy (Pagrus pagrus) along the southeast USA Atlantic coast: inferences from fishery-independent trap and video monitoring

PONE-D-23-04623R1

Dear Dr. Bacheler,

We’re pleased to inform you that your manuscript has been judged scientifically suitable for publication and will be formally accepted for publication once it meets all outstanding technical requirements.

Kind regards,

John A. B. Claydon, Ph.D.

Academic Editor

PLOS ONE
---

## [Editor Report · Acceptance letter]

26 Jun 2023

PONE-D-23-04623R1 

Low recruitment drives the decline of red porgy (*Pagrus pagrus*) along the southeast USA Atlantic coast: inferences from fishery-independent trap and video monitoring 

Dear Dr. Bacheler:

I'm pleased to inform you that your manuscript has been deemed suitable for publication in PLOS ONE. Congratulations! Your manuscript is now with our production department. 

Kind regards, 

on behalf of

Dr. John A. B. Claydon 

Academic Editor

PLOS ONE